# LOCAL-FORWARD: TOWARDS BIOLOGICAL PLAUSIBILITY IN DEEP REINFORCEMENT LEARNING

## ABSTRACT

A lasting critique of deep learning as a model for biological intelligence and learning is the biological implausibility of backpropagation. Backpropagation requires caching local outputs and propagating a global error via derivatives, neither of which are known to be implemented by biological neurons. In reinforcement learning, building more biologically plausible agents would allow us to better model human cognition and social behavior, and improve computational efficiency. We propose Local-Forward, a new temporal-difference learning algorithm (and associated architecture) that trains neural networks to predict Q-values. Rather than backpropagating error derivatives, we rely on updates that are local to each layer of the architecture and additionally use forward connections in time to pass information from upper layers to lower layers via activations. Our approach builds on the recently proposed Forward-Forward algorithm, as well as recurrence and attention in neural architectures. This approach no longer suffer the aforementioned contradictions with biology. Furthermore, as a proof-of-concept, we train reinforcement learning agents with Local-Forward to solve control tasks in the MinAtar environments, and show that our method's potential warrants further investigation because it opens avenues for more computational efficient training.

## 1 INTRODUCTION

Computer science and neuroscience have long had a reciprocal and mutually beneficial relationship. Indeed, artificial neural networks were inspired by biologically neural networks and foundational RL algorithms have close analogues to neurological processes [31]. Although artificial neural networks share many similarities to biological neural networks, they also differ in many potentially important ways. One such difference is how training occurs. Biological neural networks are trained through activation differences [18]. Instead, backpropagation [29] has been the gold standard technique for training artificial neural networks. Yet, backpropagation computes derivatives of the model prediction errors via automatic differentiation to update the model parameters and requires backwards connections in the computational graph, which contradicts known biology.

A recent proposal introduces a more biologically-plausible alternative to training artificial neural networks – the Forward-Forward algorithm [13]. In training algorithm, each layer of a neural network is greedily trained to separate positive and negative data using only the activations of that layer. This local, layer-wise loss still enables layers to learn useful representations for subsequent layers; Forward-Forward achieves comparable performance to backpropagation at training a fully-connected network on MNIST [8] and CIFAR-10 [17].

We propose a novel algorithm, Local-Forward, which builds on the Forward-Forward method and applies it to reinforcement learning [34]. Because Forward-Forward computes local updates rather than error derivatives (as would be the case in backpropagation), the approach has several potential advantages. First, it may more readily model human cognition. This has implications for areas of reinforcement learning; for instance, it allows us to better model interactions between humans and MARL-agent based models. Second, humans are able to solve many tasks that remain out of reach of artificial neural networks. Hence, biological plausibility has the potential to uncover new techniques for training and deploying reinforcement learning agents. Third, limitations of backpropagation make it computationally-expensive to scale training to increasingly larger models, or take advantage of more efficient neuromorphic hardware [15]. An algorithm like Forward-Forward may lead to significantly

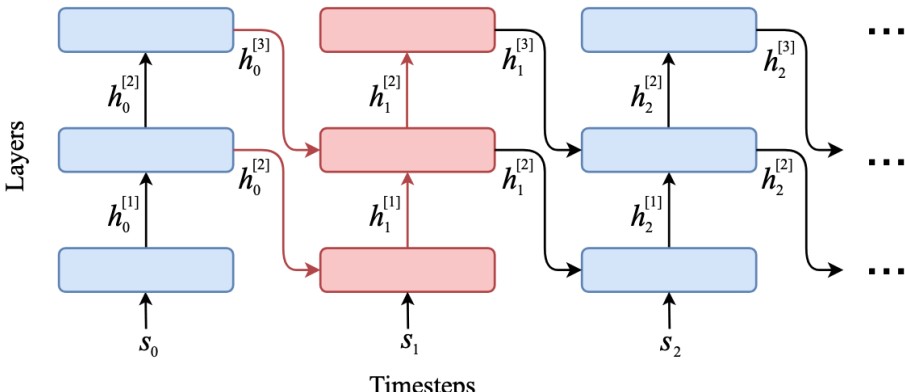

Figure 1: Network architecture of a 3-layer Local-Forward network. $h_t^{[l]}$ represents the activations of layer $l$ at time $t$, and $s_t$ the input state. The blocks are Local-Forward cells, whose inner workings are shown in Figure 2. Unlike backpropagation, loss signals do not propagate across layers or timesteps; each layer computes its own loss. To relay information, upper layers send their activations to lower layers in the next timestep. For example, red shows all active connections at $t = 1$.

more efficient computational graphs for training larger models. As reinforcement learning can be very compute intense, this is highly desirable.

Local-Forward builds on two ideas from Forward-Forward to design a novel architecture and training algorithm for deep Q-learning. The first is the local losses to replace the backpropagation of a global error, and the second is the forward [1] connections in time for sending information to lower layers; hence, the name Local-Forward. Figure 1 outlines the architecture of a Local-Forward network, unfolded in time. The structure of the network is similar to a recurrent network, except the forward connections are downwards, and each layer is a Local-Forward cell (see Figure 2), which contains additional components for computing the local Q-prediction and TD loss.

To test the capabilities of Local-Forward, we evaluate it in the MinAtar testbed [38], which are miniaturized implementations of Atari games, and compare its performance against a Deep Q Network (DQN) [24] trained via backpropagation. The MinAtar environments reflects many of the challenges faced in deep RL [4], while remaining sufficiently tractable that a simpler architecture like a fully-connected double DQN can reach good performance, making it ideal for exploring an alternative to the well-established backpropagation. Our results in Figure 3 show that Local-Forward achieves close performance to DQN in 4 of the 5 environments, and outperforms the DQN baseline in one of the environments. To further demonstrate that our algorithm is applicable across environments, we also performed experiments on several classic RL control environments, as well as MNIST and CIFAR-10, and show the results in Appendix C and Appendix D.

To summarize our core contributions:

- We propose an alternative to backpropagation for reinforcement learning that suffer less contradictions with known biology.

- We propose a new architecture and training scheme for Q-learning that builds upon Forward-Forward. Our algorithm trains a neural network that minimizes TD error using only local updates, with no gradients propagating between layers. To send information to lower layers, upper layers utilize connections that send activations forward in time.

- We empirically show that Local-Forward achieves comparable performance with a similar-sized DQN in our experiments on the MinAtar environments. We further show that the forward connections are critical to performance. Our proof of concept demonstrate the learning potential of our Local-Forward approach and architecture.

---

[1]"Forward" and "backward" are widely used both to describe direction in time and in the order of layers. This can be confusing. For the remainder of this paper, we use "forward/backward" when describing time, and "upper/lower" when describing position among layers.

## 2 BACKGROUND

### 2.1 TEMPORAL DIFFERENCE LEARNING

Formally, we consider a standard discounted infinite horizon Markov Decision Process (MDP) setting with states $\mathcal{S}$, actions $\mathcal{A}$, a transition kernel $p(s'|s,a)$, a reward function $r : \mathcal{S} \times \mathcal{A} \to \mathbb{R}$ and a discount factor $\gamma \in [0,1]$. The goal of learning is to obtain a policy $\pi(a|s)$ that maximizes the discounted future sum of rewards. The value function $Q : \mathcal{S} \times \mathcal{A} \to \mathbb{R}$ measures how valuable a given action $a \in \mathcal{A}$ is in a given state $s \in \mathcal{S}$, and can be used to directly compute a policy. It is defined via a recursive formula $Q(s,a) = r(s,a) + \gamma \sum_{s'} p(s'|s,a) \sum_{a'} \pi(a'|s')Q(s',a')$ which means classic supervised learning methods are insufficient to directly estimate it.

Temporal difference learning (TD learning) methods are a well established family of approaches that aim to solve the problem of estimating the recursively defined Q function [33]. In their most basic form, they can be implemented as lookup tables, where an estimate of the Q function is kept for each state-action pair. When experience in the form of $s, a, r, s'$ tuples becomes available from interaction with the MDP, the Q function estimates are updated using the TD error $\delta(s,a,s') = r(s,a) + \gamma \sum_{a'} \pi(a'|s')Q(s',a') - Q(s,a)$. The update rule is $Q_{k+1}(s,a) \leftarrow Q_k(s,a) + \alpha_k \delta(s,a,s')$.

In many practical applications, especially real-world interaction like those the human brain is capable of dealing with, a tabular representation of the Q values for all distinct state-action pairs is computationally infeasible. In these cases, function approximation, for example via linear regression or neural networks, is necessary to obtain an estimate of the Q function. These methods, called Fitted Q Iteration [1] or Deep Q Learning [23], use the squared temporal difference error as a regression loss $\mathcal{L}(s,a,s') = \delta(s,a,s')^2$ and update a parametric Q function approximation via gradient descent. To prevent the double sampling bias and other instabilities, only $Q(s,a)$ is updated and the next states value is estimated with an independent copy $\bar{Q}$ that is not updated. This is commonly called the bootstrapped Q loss and can be written down as:

$$\mathcal{L}(s,a,s',\theta) = \left( Q_\theta(s,a) - \left[ r(s,a) + \gamma \max_{a'} \bar{Q}_\theta(s',a') \right] \right)^2 .$$

We are particularly interested in exploring methods to progress the biological plausibility of TD learning algorithms, because TD learning is widely studied as an explanatory model in neuroscience for biological learning [32; 6]. Empirical evidence suggests that the phasic activity of dopamine neurons in the midbrain correlates with prediction errors, which can be formalized via TD, and provides a global learning signal for synaptic modification [10].

### 2.2 FORWARD-FORWARD LEARNING

The Forward-Forward (FF) algorithm [13] is a greedy multi-layer learning algorithm that replaces the forward and backward passes of backpropagation with two identical forward passes, positive and negative. These passes differ only in the data fed through the network and the objective they optimize. For a positive pass, the update increases the goodness $g$ of that layer. In the case of a negative pass, the update decreases goodness. While Hinton [13] proposes several measures of goodness, the de facto measure used is the sum of each squared hidden activation $h_i$ of the layer minus a threshold $\theta$, i.e. $g = \sum_i h_i^2 - \theta$. The positive data are real examples while the negative data is 'fake.' One way to generate the fake data is by masking and overlaying real examples. For the MNIST dataset, this could be combining the top half of a 7 with the bottom of a 6. The goal of the model is to correctly classify an input as either positive (real) or negative (fake). To optimize each layer's weights, FF minimizes the binary cross-entropy loss between the sigmoid of the goodness $\sigma(g)$ and the ground truth.

Each of the positive and negative pass performs a local update on every layer. Whereas backpropagation is a global algorithm that requires the entire network to be updated at once, the FF is a local algorithm that updates each layer independently through the use of layer-wise local losses. For a feed-forward architecture, one glaring issue with this approach is that representations are learned one layer at a time with no way for later layers to influence earlier ones. One way to address this is to use a multi-layer recurrent architecture, where for each layer $\ell$ in the network, the input is determined by output of the layers above $\ell + 1$ and below $\ell - 1$ at the previous time step $t - 1$. This allows the network to learn representations in a greedy fashion, but also allows for top-to-bottom information flow throughout the network.

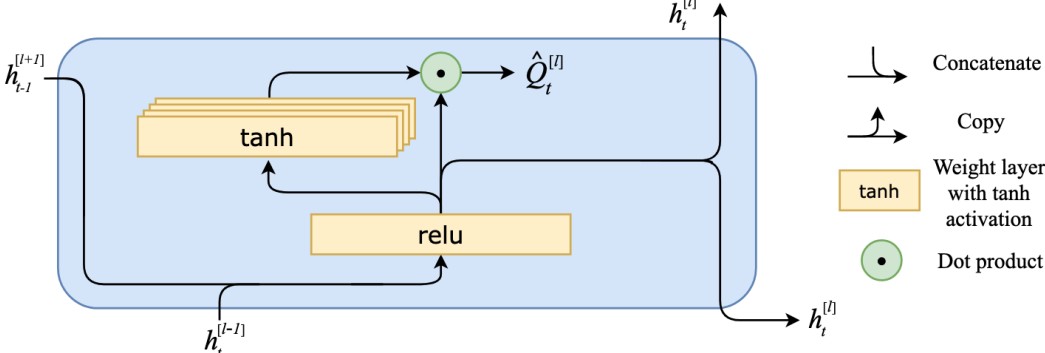

Figure 2: Inner workings of our proposed Local-Forward cell (i.e., hidden layer). $h_t^{[l]}$ is the activations of the cell $l$ at time $t$, and $\hat{Q}_t^{[l]}$ is the Q-value prediction. We compute the cell's activations $h_t^{[l]}$ using a `ReLU` weight layer, then use an attention-like mechanism to compute $\hat{Q}_t^{[l]}$. Specifically, we obtain $\hat{Q}_t^{[l]}$ by having the cell's `tanh` weight layers, one for each action, compute attention weights that are then applied to $h_t^{[l]}$. Each cell computes its own error.

## 3 METHOD

The core purpose of the method is to obtain a TD-learning algorithm that works via purely local updates; we do not propagate any loss signals between layers of the network. To achieve this, the network is composed of individual layers that each solve the TD regression problem individually. We refer to each of these layers as a *cell*. The cells are connected via their activations, which they provide as inputs to the next cell (i.e., layer). The whole architecture is summarized in Figure 1.

### 3.1 CELL INTERNALS

Our network architecture is designed around individual cells which perform local computations with the squared TD loss. Unlike FF, which separates positive from negative data during training, a binary classification task, predicting the Q-value is a regression task and requires more precision. Inside the Local-Forward cells, we use an attention-like mechanism [2] that learns a weighted sum of the layer's hidden activations to predict the Q-value. The hidden activations are passed to other cells, but the attention weights and Q-value are withheld.

The concept of a cell is reminiscent of the design of a recurrent neural network. In our case, there is a single cell per layer – so we may use the terms interchangeably for clarity of exposition. The vital difference is that no loss information is propagated between cells via backpropagation, i.e. there is no backpropagation in time, and the same activations are passed to the cell (i.e., hidden layer) above, which happens in the same timestep, and the layer below, at the next timestep. In the absence of backpropagation, these activations provide a pathway for upper cells to communicate information to lower cells. We discuss in more detail how these connections operate in Section 3.2.

The cell design is presented in Figure 2. Recall that each cell corresponds to a 'layer' in a typical network architecture. Each cell is mechanistically identical and takes in two inputs: the hidden activations of the cell below at the current timestep (or the observation if at the lowest cell), and the hidden activations of the cell above at the previous timestep. They produce two identical outputs, sent to the cell above immediately, and the cell below at the next time step. At the start of an episode, the activations from the previous timestep are zero. The top cell only receives one input and produces one output. Our design currently only admits one-dimensional representations. Hence, we leave extensions to convolutional 2D and 3D networks to future work.

We introduce an attention mechanism [2] to compute the Q-value prediction. Each cell first computes its hidden activation, $h_t^{[l]}$, using the layer above's hidden activations from the previous timestep, $h_{t-1}^{[l+1]}$, and the layer below's hidden activations at the current timestep, $h_t^{[l-1]}$. Specifically, it passes

the concatenation of $h_{t-1}^{[l+1]}$ and $h_t^{[l-1]}$ through a `ReLU` weight layer (shown in Figure 2) to get $h_t^{[l]}$. The `ReLU` weight layer multiplies the concatenated input $[h_t^{[l-1]}, h_{t-1}^{[l+1]}]$ by its learned weight matrix $W_{in}^{[l]}$, then applies the `ReLU` nonlinearity function. Next, the cell passes $h_t^{[l]}$ through the `tanh` weight layers, each corresponding to an action $a$, which multiplies $h_t^{[l]}$ by its learned weight matrix $W_{query}^{[l]}$, then applies the tanh function. The output of the `tanh` layers plays a role akin to attention weights, although they do not have to sum to 1. Finally, the cell takes the inner product between the output of the `tanh` layers, $\tanh(W_{query}^{[l]} \cdot h_t^{[l]})$, and the hidden activations $h_t^{[l]}$, to compute the cell's Q-value prediction $\hat{Q}^{[l]}(s_t, a)$. This structure is heavily inspired by the successful attention architecture [2], and allows each cell to attend to the incoming information dynamically per action.

Again, recall that each cell corresponds to a layer of our neural network. Each cell reuses its internal weights over time, meaning for instance that the matrix $W_{in}^{[1]}$ is used at each timestep in the first cell. However, across cells, weight matrices are independent, meaning $W_{in}^{[1]} \neq W_{in}^{[2]}$.

To summarize, the full computation performed by the cell is:

$$h_t^{[0]} = s_t, \quad h_t^{[L+1]} = 0, \quad h_{-1}^{[l]} = 0$$
$$h_t^{[l]} = \text{relu}\left(W_{in}^{[l]} \cdot \left[h_t^{[l-1]}, h_{t-1}^{[l+1]}\right]\right)$$
$$\hat{Q}^{[l]}(s_t, a) = \tanh\left(W_{query}^{[l]} \cdot h_t^{[l]}\right)^{\mathsf{T}} h_t^{[l]}$$

### 3.2 NETWORK CONNECTIONS

As shown in Figure 1, each cell passes its state to the cell $l + 1$ above at the current timestep $t$, and to the cell $l - 1$ below in the next timestep $t + 1$. The information flow is strictly uni-directional to account for the causal sequence of RL environments. This is necessary as interacting with the environment happens sequentially, meaning future information will not be available when acting.

Although we do not backpropagate gradients across cells, information does flow from upper layers to lower layers via the temporal connection. The upper layers use the connections to communicate with lower layers via activations, which is more biologically plausible [25]. Our results in Section 3 suggest that these connections can greatly increase network performance in the absence of backpropagation.

### 3.3 LEARNING ALGORITHM

We formally describe our learning algorithm in Algorithm 1.

Each cell is updated using the Double Q-learning algorithm [36]. This is a simple and robust extension to the Q-learning algorithm [23] presented in Section 2. In Double Q-learning, the maximization problem defined to find the optimal next state action is solved over the online network, but the evaluation of that action uses a separate target network. This alleviates overestimation problems commonly plaguing deep Q-learning, as different networks are used to estimate and evaluate the optimal action.

The loss itself needs not change to be adaptable to our setup. The target network is produced, as in double DQN implementations, by replicating the proposed network structure per cell and periodically copying the learned parameters. Different from standard deep Q-learning however, each layer of the network computes its own local loss and update using a global reward signal. Specifically, the reward is broadcasted to all cells, and each cell computes their own Q-value prediction, TD-target, and TD-loss for learning. Globally broadcasting the reward signal to enable local learning is biologically motivated, imitating reward-learning processes in the striatum, which globally broadcasts reward signals via dopamine release that induces local updates [19] (see Appendix E for more details). During inference, the network uses the average Q-values across the layers to produce a final prediction. This can also be interpreted as a bagging method, in which several weaker regression models work jointly to compute a prediction.

During training, we use a standard replay buffer which stores every transition observed from the environment. We replay sequences of short length, similar to how other recurrent architectures are trained, and compute the local updates for each cell sequentially according to the network graph.

---

**Algorithm 1** Local-Forward Q-Learning

---

Initialize replay memory $\mathcal{D}$ to capacity $\mathcal{N}$
Initialize action-value functions $Q^{[1]}, \ldots, Q^{[L]}$ with random weights $\theta^{[1]}, \ldots, \theta^{[L]}$
Initialize target function $\hat{Q}^{[1]}, \ldots, \hat{Q}^{[L]}$ with weights $\theta^{[i]'} = \theta^{[i]}$ for every $i \in \{1, \ldots, L\}$
**for** episode 1, $M$ **do**
    Initialize $s_1 = $ initial state
    **for** $t = 1, T$ **do**
        With probability $\varepsilon$ select a random action $a_t$
        otherwise select $a_t \in \arg\max_a \frac{1}{L} \sum_{l=1}^{L} Q^{[l]}(s_t, a; \theta^{[l]})$
        Execute action $a_t$ in emulator and observe reward $r_{t+1}$ and next state $s_{t+1}$
        Store experience $(s_t, a_t, r_{t+1}, s_{t+1})$ in $\mathcal{D}$
        Sample minibatch of $K$-sized episodic sub-trajectories

$$(s_{j-K}, a_{j-K}, r_{j-K+1}, s_{j-K+1}), \ldots, (s_j, a_j, r_{j+1}, s_{j+1})$$

        from $\mathcal{D}$ with uniform randomness
        Replay states $s_{j-K}, \ldots, s_{j-1}$ through online network to obtain layer activations $h_{j-1}^{[1]}, \ldots, h_{j-1}^{[L]}$
        Set current layer input $x \leftarrow s_j$
        **for** layer $l = 1, \ldots, L$ **do**
            Set TD target $Y_j^{[l]} = \begin{cases} r_{j+1} \text{ if episode terminates at step } j+1 \\ r_{j+1} + \gamma \max_a \hat{Q}^{[l]}\left(s_{j+1}, Q^{[l]}(s_{j+1}, a; \theta^{[l]}); \theta^{[l]'}\right) \text{ otherwise} \end{cases}$
            Set prediction $y_j \leftarrow Q^{[l]}(s_t, a_j; \theta^{[l]})$ using $x$ and last activations $h_t^{[l-1]}$ and $h_{t-1}^{[l+1]}$
            Perform a gradient descent step on $(Y_j^{[l]} - y_j)^2$ with respect to the layer parameters $\theta^{[l]}$
            Set input for next layer $x \leftarrow h_j^{[\ell]}(x)$
        **end for**
        Every $C$ steps reset $\theta^{[i]'} = \theta^{[i]}$ for every $i \in \{1, \ldots, L\}$
    **end for**
**end for**

---

To use the network for environment interaction, we simply keep the recurrent state for each cell and run the computation again, choosing the maximum action according to the aggregated Q function at each timestep. We also found it valuable to use $\epsilon$-greedy exploration to obtain more diverse data.

## 4 EXPERIMENTS

As our main purpose is to demonstrate the potential of a novel architecture and learning algorithm, rather than achieve state-of-the-art performance, we opt for a simple implementation with few extensions. Our experiments are designed to support our claim that Local-Forward architecture and algorithm is a promising alternative to backpropagation – and addresses the latter's biological implausibility. In particular, we empirically show that Local-Forward cells are able to learn meaningful representations by optimizing their local TD error. This setup is similar to the evaluation presented by Hinton [13]. However, we do not use a convolutional network, and the only extension we employ is Double Q-learning [36] to improve learning stability.

**Environments.** We run our experiments in the MinAtar [38] environment suite. MinAtar is an simplified implementation of 5 Atari 2600 games: Seaquest, Breakout, Asterix, Freeway, and Space Invaders. We use version 1 for all environments. The input frame size is $10 \times 10$, and the $n$ different objects in each game are placed on separate frames, resulting in a $10 \times 10 \times n$ input observation. All tasks have discrete action spaces with up to 6 actions.

We select this set of environments because they reflect challenges in solving control tasks with high-dimensional observations [4], where neural networks are widely adopted, while still remaining simple enough that convolutional layers are not critical. Although the DQN baselines published by [38]

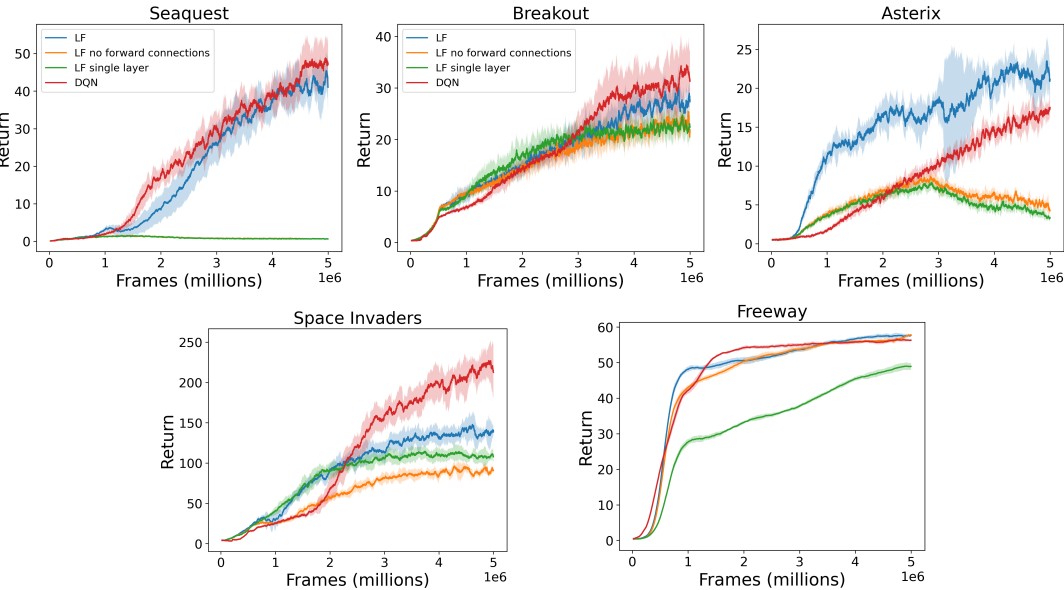

Figure 3: Episodic returns of Local-Forward, DQN, Local-Forward without forward connections, and single layer Local-Forward cells on the MinAtar environments. Lines show the mean episode reward over 10 seeds and the shaded area conforms to 3 standard errors of the mean.

are obtained using a convolutional neural network (CNN), a larger DQN with only fully-connected layers can solve all tasks with comparable or stronger performance.

**DQN baseline.** We compare our results against a fully-connected DQN to make performance comparisons more direct and informative. As the original baselines by Young & Tian [38] used a CNN, for fairness of comparison, we replaced the CNN with fully-connected layers, and tuned hyper-parameters for the new architecture. We find that the new architecture is consistently as good or better than the one presented by Young & Tian [38]. In particular, we use a double DQN with 3 hidden layers of 1024, 512, and 512 hidden units with `ReLU` activations. This baseline achieves strong performance and has a similar number of trainable parameters as our networks.

**Network architecture and hyperparameters.** We use a 3 hidden layer network with forward activation connections. The cell output sizes are 400, 200 and 200. Because we have additional connections within cells, and from upper cells to lower cells, this yields a similar number of trainable parameters as the DQN (see Appendix B). To study the performance effects of the forward connections to lower layers, we also run experiments using the same network without these connections.

We train agents in each environment for 5 million time steps, with a data-to-update ratio of 4. We use a learning rate of $10^{-4}$, Adam optimization with a warmup scheduler and annealing, an exploration fraction of 0.1, and $\epsilon$-greedy exploration that anneals from 1.0 to 0.01. For more details, see Appendix A. We use the same network and hyperparameters for all 5 environments to test the robustness of our architecture and learning algorithm across similar environments.

## 5 RESULTS

We present the results of Local-Forward on MinAtar environments in Figure 3, and compare its performance against DQN, Local-Forward without the forward connections to lower layers, and a single layer Local-Forward cell.

**Comparison against DQN baseline.** We find that the Local-Forward algorithm is able to learn stable policies reliably in all test environments. Our agent achieves comparable performance to DQN on MinAtar's Breakout, Asterix and Freeway, slightly surpassing DQN's performance on Asterix and

Freeway, while DQN shows stronger performance in Seaquest and Space Invaders. Our findings here are slightly surprising as our DQN baseline strongly outperforms the results presented by Young & Tian [38], which leads us to believe that the exact network architecture and hyper-parameter tuning method might be important to consider for future experiments with this benchmark suite.

However, it is promising that Local-Forward matches or outperforms benchmark DQN MinAtar results presented by both Young & Tian [38] and [4], which leads us to believe that the method can be a viable alternative approach to backpropagation in deep reinforcement learning if further explored.

**Forward connections to lower layers.** To further measure the performance impact of the forward connections in time, we compared the temporal forward version of the network with the performance of a 3-layer Local network using only upward connections and no forward connections. All other hyper-parameters are the same as the recurrent 3-layer Local-Forward network. This results in a moderate drop in performance in SpaceInvaders and Breakout, but devastating drops in Seaquest and Asterix. This strongly suggests that the information channels from the upper to lower layers is vital for performance for many tasks. On this same note, Local-Forward's ability to achieve similar performance to DQN when the forward connections are added suggest that forward connections may be an effective replacement for backpropagation in certain environments.

In Seaquest, where the performance gap is most apparent, the agent's struggle may be attributed to the failure of learning the resurfacing mechanism, which traps it at a local minima. In this environment, the agent must regularly resurface while carrying a diver to replenish oxygen. In the short term, the agent can acquire more reward if it prioritized attacking enemies, rather than managing oxygen, but if it runs out of oxygen the episode ends. We observed that the Local-Forward agent without forward connections and the single layer agent both struggled to manage oxygen, often resurfacing randomly or without first acquiring a diver. In contrast, the Local-Forward agent with the forward connections learns to manage their oxygen more optimally, and often sacrifice short-term rewards to maintain higher oxygen. In fact, during some trials the Local-Forward agent became ultraconservative with oxygen, which led to the agent surviving on average over 1500 timesteps per episode over several thousand episodes before it changed strategy. Interestingly, this same behavior was not observed in DQN agents, whose average episode lengths are consistently under 800. This suggests that Local-Forward has the potential to learn completely different policies from DQN.

Similarly, in Asterix, we observed that the agent without forward connections struggles to adapt to increased difficulties as the game progresses and the enemies increase. As training progresses, the agent's performance decreases as it unsuccessfully tries to find a strategy for later stages of the game.

However, in Freeway the forward connections seem to have minimal impact on the Local-Forward agent's performance. This is likely a result of the relative simplicity of the environment, and its fixed-episode length format. Unlike the other environments, the episode length of Freeway is fixed at 2500 timesteps, and agents receive reward each time they successfully cross a road during the episode. Once the agent nears an episodic return of 60, it cannot further increase performance, because there is no time to earn more reward even with a near optimal policy. This is reflected in the steep increases in returns in the early phase of learning, followed by rapid flattening. Therefore, both the DQN and Local-Forward agent has no room to pull ahead in performance from the others.

**Single layer performance.** Another important question about the presented architecture is whether the majority of the learning is accomplished by the lowest cell which is directly supplied with the state. In that case, the inner attention mechanism of the cell can function like a shallow, two layer fully-connected network, which can perform well in simpler RL environments.

Our ablation study of a single layered cell shows that it is insufficient to achieve strong performance on any of the benchmarking tasks. In all environments, including the simpler Freeway, there are notable gaps between the performance of the single layered cell and the 3-layer Local-Forward network, showing that the multi-layer architecture with forward connections provide crucial performance benefits.

Additionally, like the Local-Forward network without forward connection, the single layer cell also struggles at Seaquest and Asterix, while the Local-Forward network with the connections thrive. This also provides circumstantial evidence to support the hypothesis that both the multi-layered architecture and the forward connections are essential, and that the forward connections can convey meaningful information to lower layers directly via activations for coordinating learning.

## 6 LIMITATIONS

We make progress towards biologically plausible deep RL by replacing backpropagation with local-only losses and forward connections in time. The latter allow us to pass information to lower layers. However, our model does not capture all aspects of biological neural networks.

For example, biological neural networks have backward connections that allow for hierarchical predictive coding and the updating of weights through the comparison of pre- and post- synaptic weights [5]. The additional loss function that captures this comparison minimizes the difference between predicted and actual activations. Instead, the model we propose analyzes local losses based on TD errors alone. Additional work will be required to consider other aspects of biological cognition, such as Hebbian learning [12]. Doing so may affect local losses. For instance, a tension between the Hebbian (contrastive) loss and error driven loss could arise within different components of the architecture, which would make it difficult to scale to larger models. Careful consideration will thus be required as to the choice of specific error that each component is sensitive to.

While we do not propagate any loss signal between cells, within each cell we use an attention-like mechanism [2] to compute the Q-value predictions. This attention-like mechanism is trained via backpropagation, albeit we only propagate through one layer: the loss signal for the `ReLU` weight layer is propagated through the `tanh` weight layer. Given that the architecture of this mechanism is very simple, it is likely possible to replace backpropagation as the training algorithm for this mechanism with a more biologically plausible alternative, such as Difference Target Propagation [9] or Equilibrium Propagation [30]. We plan to explore this direction in future work.

We present this work as an alternative pathway into finding learning architectures and algorithms. In this sense, the paper serves a both an investigation into components that are closer to biological structures, as well as a proof-of-concept that shows this learning paradigm can perform at least as well as established algorithms as DQN. We do not claim state-of-the-art performance over more recent algorithms, as this is a radically departure from established algorithms and more time would be needed to match the performance of algorithms like Rainbow or MuZero. But we believe that our results show that this alternative is feasible and promising.

## 7 RELATED WORK

While many early RL algorithms are strongly inspired by neuroscience [37; 7], the machine learning literature has since diverged from these roots and focused strongly on combining reinforcement learning with modern neural networks to produce competitive algorithms [22; 21; 11] for benchmark suits like the Arcade Learning Environment [3] or the DeepMind Control Suite [35].

Biologically inspired learning algorithms seek to reconnect machine learning with neurobiology. Similar to our work, [28] proposes an alternative to backpropagation for deep Q-learning that is more biologically plausible, based on the Neural Generative Coding framework [27]. Other related approaches for more plausible deep learning include Feedback Alignment [20], which addresses the weight transport problem of backpropagation [20], target propagation [18], which generates local errors, and SoftHebb [16], which uses Hebbian learning.

## 8 CONCLUSION

We proposed Local-Forward, a deep Q-learning algorithm and architecture that alleviates several contradictions between deep RL and biological learning, particularly due to the biological implausibility of backpropagation. Each Local-Forward cell computes its own local TD error, based on the difference between its prediction and the TD target. The TD target is provided to all cells; in parts of the brain this could be achieved by a global signal via dopamine neurons. To send information to lower layers, upper layers send their activations forward in time to the layer below, hence the name Local-Forward. We empirically showed that Local-Forward can learn control tasks in the MinAtar environments, and even surpasses a similar-sized DQN in performance in 2 environments. Thus, we believe it warrants further investigation. We outline some areas for investigating potential improvements in our limitations.

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

## A    MODEL HYPERPARAMETERS

We show the hyperparameters of the Local Forward networks and DQN used in our MinAtar experiments.

| Hyperparameter | Local Forward | DQN |
|---|---|---|
| Learning rate | $10^{-4}$ | $10^{-4}$ |
| Exploration fraction | 0.1 | 0.1 |
| Final epsilon at end of exploration | 0.01 | 0.01 |
| Loss function | Mean squared error | Mean squared error |
| Max gradient norm | 1.0 | 1.0 |
| Batch size | 512 | 512 |
| Gamma | 0.99 | 0.99 |
| Training frequency | 4 | 4 |
| Replay buffer size | $5 \times 10^6$ | $5 \times 10^6$ |
| Target network update frequency | 1000 | 1000 |

We also use a learning rate scheduler for both networks, which linearly increase learning rate to $10^{-4}$ in 500000 steps, then cosine decay until $3 \times 10^{-5}$ over remaining training steps. These hyperparameters were jointly tuned and shared by both the baseline DQN and Local-Forward network.

## B    NUMBER OF TRAINABLE PARAMETERS

In Section 4, we stated that we used a 3-layer Local-Forward network with output sizes 400, 200 and 200 respectively, and that we chose these sizes to yield a similar number of trainable parameters as the DQN with 1024, 512 and 512 units for fairer comparison. Here we provide the detailed calculations.

Since each MinAtar environment differs slightly in number of input channels and size of action space, the the number of parameters also vary slightly across environments. Assuming an environment with 4 input channels and 3 actions, the number of trainable parameters for the DQN is $10 \times 10 \times 4$ (input size) $\times 1024$ (first layer weights) $+1024$ (first layer biases) $+1024 \times 512$ (second layer weights, etc.) $+512 + 512 \times 512 + 512 + 512 \times 3$ (actions) $+3 = 1,199,619$.

The number of trainable parameters for the Local-Forward (LF) network is $(10 \times 10 \times 4 + 200)$ (input size + size of hidden activations of second layer from last time step) $\times 400$ (first layer LF cell `ReLU` weight layer weights) $+400$ (first layer LF cell `ReLU` weight layer biases) $+3 \times 400 \times 400$ (first layer LF cell `tanh` weight layer weights) $+3 \times 400$ (first layer LF cell `tanh` weight layer biases) $+(400 + 200)$ (first layer output size + size of hidden activations of third layer from last time step) $\times 200$ (second layer LF cell `ReLU` weight layer weights, etc.) $+200 + 3 \times 200 \times 200 + 3 \times 400 + 200 \times 200 + 200 + 3 \times 200 \times 200 + 3 \times 200 = 1,123,200$.

## C    EXPERIMENTS ON CLASSIC CONTROL ENVIRONMENTS

Beyond the MinAtar testbed, we also evaluate Local-Forward on 4 classic control environments: Cart Pole (also known as inverted pendulum), Mountain Car, Lunar Lander, and Acrobot, and compare its performance against DQN. We show our results in Figure 4.

The main purpose of these experiments is to further support Local-Forward's robustness across different RL environments, and indirectly compare its performance against Active Neural Generative Coding (ANGC) [28], another biologically-inspired deep RL algorithm that provides a more plausible alternative to backpropagation. [28] evaluates ANGC in 4 environments: Cart Pole, Mountain Car, Lunar Lander, and a custom robot-arm-reaching environment. As [28] has yet to publicly release the code for their ANGC agent, we are unable to directly evaluate ANGC on the MinAtar testbed to compare performance; thus, we instead evaluate Local-Forward on ANGC's environments.

In addition, we implement 2 DQNs with different hyperparameters as baselines for comparison: first is CleanRL's reference DQN for classic control problems [14], and second is the DQN tuned by [28] for comparison against their ANGC agent, which we refer to as ANGC's DQN. The CleanRL DQN

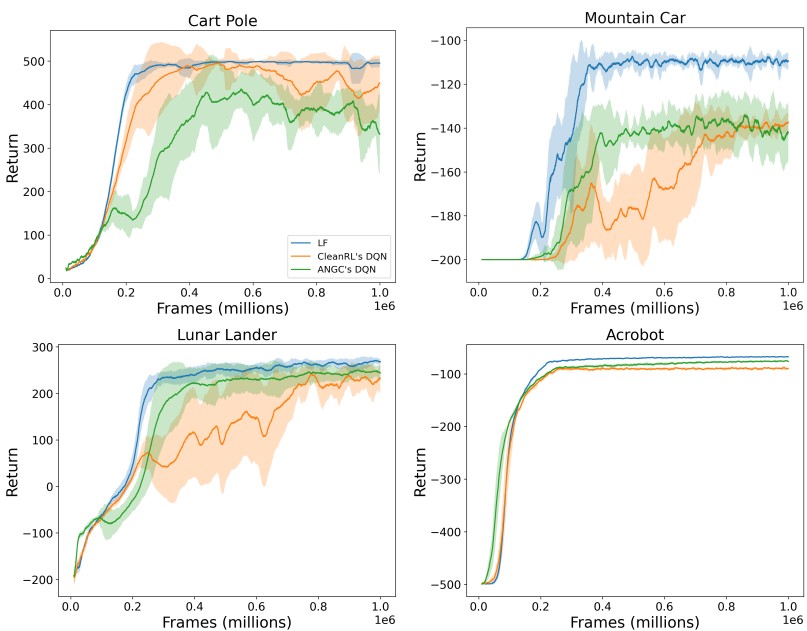

Figure 4: Episodic returns of Local-Forward, CleanRL's DQN, and ANGC's DQN. Lines show the mean episode reward over 10 seeds and the shaded area conforms to 3 standard errors of the mean.

provides a well-established, publicly-vetted baseline, whereas ANGC's DQN serves as a reference point for us to indirectly compare Local-Forward's performance with ANGC.

To better demonstrate Local-Forward's robustness across environments, we use the same set of hyperparameters for each of the environments. Specifically, we use a 2-layer Local-Forward network, with output sizes 128 and 96, a fixed learning rate of $2.5e^{-4}$ with Adam optimization, an exploration fraction of 0.2, and final epsilon of 0.05. All other hyperparameters are the same as our network for MinAtar, shown in Appendix A. As neither CleanRL nor ANGC's DQN uses a learning rate scheduler, we also removed ours for better comparison. CleanRL's DQN is a 2-layer network with 120 and 84 units; the same DQN is used for all 4 environments. ANGC uses different 2-layer DQNs for each of their environments, which they tuned for environment-specific performance. Their Cart Pole DQN has 256, 256 hidden units; Mountain Car has 256, 192, and Lunar Lander 512, 384. For Acrobot, we used the same hyperparameters as the Cart Pole DQN, given the similarity of the environments, which worked well. For more hyperparameter details, we refer the reader to [14] and [28]; both works present excellent detail.

We made two adjustments when reproducing CleanRL and ANGC's DQNs. First, since Local-Forward utilizes double Q-learning to improve its learning stability, for better comparison, we also enhanced both DQNs with double Q-learning, given that it is a simple enhancement without additional computational costs. Second, specific to ANGC's Cart Pole DQN, we found that the agent's learning has very high variance with the given hyperparameters, and the agent does not consistently achieve high performance. To counter this, we decreased the learning rate from the provided $5e^{-4}$ to $2.5e^{-4}$, and increased the target network update frequency (referred to as $C$ in [28]) from 128 to 500, which is CleanRL's value. This improved both the agent's stability and average return.

Our results in Figure 4 show that Local-Forward achieves strong performance across all 4 tasks, learning more stably and achieving higher average episodic return than the DQNs in all environments. Most notably, in comparison to DQN, Local-Forward more consistently reaches the maximum 500 return in Cart Pole, and achieves significantly higher return in Mountain Car. It also surpasses DQN's performance slightly in both Lunar Lander and Acrobot. These results suggest that Local-Forward is highly competitive against ANGC in these tasks.

We are in contact with the authors of [28], and look forward to performing more direct comparisons between Local-Forward and ANGC once their code is publicly released. As ANGC and Local-

| Dataset | Hidden activations | Layer 1 | Layer 2 | Layer 3 | Layer 4 |
|---------|-------------------|---------|---------|---------|---------|
| MNIST | 500 | 0.9848 | 0.9862 | 0.9865 | 0.9867 |
| MNIST | 1000 | 0.9834 | 0.9860 | 0.9866 | 0.9870 |
| MNIST | 2000 | 0.9834 | 0.9862 | 0.9878 | 0.9874 |
| CIFAR-10 | 500 | 0.4715 | 0.5234 | 0.5394 | 0.5501 |
| CIFAR-10 | 1000 | 0.4750 | 0.5358 | 0.5496 | 0.5588 |
| CIFAR-10 | 2000 | 0.4841 | 0.5440 | 0.5610 | 0.5725 |

Table 1: Test accuracy of each layer of 4-layer Local-Forward networks trained on MNIST and CIFAR-10. Each layer has the same number of hidden activations; for example, the first row refers to a Local-Forward network with layers of 500, 500, 500, and 500 activations. Since each layer makes its own prediction, we can easily see how performance increases per layer.

Forward learn with different signals, they are not competing approaches, and may possibly be integrated to achieve greater performance– we also look forward to exploring this potential.

## D  EXPERIMENTS ON MNIST AND CIFAR-10

To further investigate the generalizability of Local-Forward across different learning tasks, we also evaluate Local-Forward on two supervised learning datasets, MNIST and CIFAR-10, and present our results here. Q-learning is a regression problem; to adapt our architecture to solve these classification problems, we simply change the loss function to cross entropy, and make each of the `tanh` weight layers correspond to a class, instead of an action. Using a 4-layer Local-Forward network with output sizes 2000, 2000, 2000 and 2000, we achieved 98.74% test accuracy on MNIST, and 57.25% test accuracy on CIFAR-10. These results are in line with the results achieved by Forward Forward in [13].

We show our results with different sized networks in Table 1. As each layer in the network makes its own prediction, we additionally show the test accuracy of every layer. Remarkably, for almost every network size, the test accuracy steadily increases per layer. Like our experiments in RL, this supports that Local-Forward cells can learn to coordinate with each other for better performance without the need to backpropagate error signals across layers.

For clarity, we want to note that these experiments are performed simply to add another perspective to evaluate the robustness of Local-Forward's learning across tasks. Local-Forward is intentionally designed for Q-learning, rather than supervised learning, as it imitates biological processes of reward-based learning.

## E  BIOLOGICAL INSPIRATION OF LOCAL-FORWARD

Here we elaborate on the connections between neurobiology and Local-Forward. Local-Forward is heavily biologically inspired. We saw a potential connection between the design principles of Forward-Forward (FF) [13] and biological reward-based learning via striatic dopamine release; Local-Forward is our attempt to synthesize the two to improve deep Q-learning.

FF provides an alternative to backpropagation that is more biologically plausible and computationally efficient, because it does not suffer from weight transport [20] or update locking [28]. Like Local-Forward, FF globally broadcasts the target to each layer of the network, which each computes their own local loss and updates. However, FF does not provide a biologically plausible explanation on how the target signal can be globally broadcasted in the brain. Interestingly, recent research in neuroscience by [19] found that in the striatum, dopamine release can be a global signal that induces local changes to reward stimulation, providing a potential answer in the case of reward-based learning. Additionally, biological reward-based learning has close ties with TD-learning, which is used as an explanatory model for dopamine-induced synaptic modifications in the striatum [32]. The alignment between FF's principle of globally-induced local learning, and evidence of biological

globally-induced local modifications for reward-based learning in the brain, is the central biological insight that motivated us to develop Local-Forward as a Q-learning algorithm.

To be clear, we do not believe that Local-Forward plausibly represents the complete reward-learning process in the striatum. Unlike backpropagation, the brain likely utilizes multiple types of learning in conjunction, both supervised and unsupervised, using local and global signals [26]. Local-Forward only models one type of learning, i.e. error-driven learning using global reward signals to induce local updates. Other biologically plausible algorithms, such as ANGC [28], may provide explanations to other types of biological learning. Indeed, aspects of Hebbian learning or active inference are likely critical to achieving a full biologically plausible, efficient, and powerful learning system. In that light, we examine Local-Forward here not to replace other methods, but to demonstrate that this one principle alone is sufficient to be able to solve some complex RL tasks nearly as well as backprop. We believe that a key to achieving general, human-like intelligence will be the integration of these different learning methods and learning signals; this is an exciting direction we aim to explore in future work.

