# OpenReview forum: "Local-Forward: Towards Biological Plausibility in Deep Reinforcement Learning"
_ICLR.cc/2024/Conference — Submitted to ICLR 2024_

### Official Review · Reviewer_AyQt · 2023-10-17

**Soundness:** 3 good
**Presentation:** 3 good
**Contribution:** 3 good
**Rating:** 6
**Confidence:** 4

**Summary:**

The paper "Local-Forward: Towards Biological Plausibility in Deep Reinforcement Learning" critiques the biological implausibility of backpropagation in deep learning. It introduces "Local-Forward", a temporal-difference learning algorithm that predicts Q-values without backpropagating error derivatives. Instead, it employs local updates and forward connections in time to convey information between layers. This approach, inspired by the Forward-Forward algorithm and attention mechanisms, aims to better model biological neural networks. The method is tested on MinAtar environments, emphasizing its potential for more efficient training and its alignment with biological processes.

----- After rebuttal -----

I appreciate the additional experimental results and clarification from the authors. I feel most of my concerns are addressed. I changed my rating from 3 to 6 accordingly.

**Strengths:**

- The paper is written clearly and smooth to follow.
- Exploring the application of Forward-Forward algorithms in various field such as deep RL is an interesting direction to explore.
- The experimental results show the effectiveness of the proposed Local-Forward algorithm, which has probably large future potential.
- The algorithm is novel to my knowledge.

**Weaknesses:**

## Major weakness

- Inadaquent disucssion about and comparison with related methods, especially backprop-free algorithm in Deep RL. See my questions below.

- Lack of depth in experimental evaluations. The paper could benefit from more comprehensive analysis and wider range of testbeds. See my questions below.


## Minor

- There is not enough background knowledge about the forward-forward algorithm. Sec. 2.2 only describes the rough idea of it, but not the detailed mathematics of how it works. This is important because it is the basis of the proposed Local-Forward algorithm.

- While the claim is "we do not propagate any loss signal between cells", actually the reward signal is used globally (Alg. 1). This should be clarified.

**Questions:**

1. There exist other backprop-free learning algorithms such as [a,b] and evolutionary algorithms like CMA-ES [c, d], which can be used for deep RL, how do your method compare with them? I believe a performance / computation cost comparison with them could consolidate the advantage of the proposed Local-Forward model.

2. What are the design motivation of the proposed Local-Forward cell (Fig.2)? The authors state that "The concept of a cell is reminiscent of the design of a rnn". However, MinAtar tasks are MDPs and can be handled by feedforward NN, why the paper tries to solve MinAtar with recurrent connections?

3. I am also curious about the performance of Local-Forward in supervised learning tasks such as MNIST and CIFAR. It does not need to be very good since the method is designed for TD-learning, but such additional experimental results may provide better understanding of the suitable scope of the proposed model.

4. What are the computational cost of Local-Forward compared with DQN? e.g., clock time, num of model parameters, FLOPS. As each layer computes Q value, is it more computationally expensive than DQN?

5. Are there any additional biological insights of the Local-Forward model rather than it can solve MinAtar? For example, how the internal representation of neurons compare with animals' neurons and DQN's neurons.


### Reference

[a] Ororbia A, Kifer D. The neural coding framework for learning generative models[J]. Nature communications, 2022, 13(1): 2064.
[b] Ororbia A G, Mali A. Backprop-free reinforcement learning with active neural generative coding[C]//Proceedings of the AAAI Conference on Artificial Intelligence. 2022, 36(1): 29-37.
[c] Hansen N, Ostermeier A. Completely derandomized self-adaptation in evolution strategies[J]. Evolutionary computation, 2001, 9(2): 159-195.
[d] Ha D, Schmidhuber J. Recurrent world models facilitate policy evolution[J]. Advances in neural information processing systems, 2018, 31.

---

> ### Author Response · Authors · 2023-11-18
>
> We thank the reviewer for their comments and suggestions.
>
> We particularly appreciate that the reviewer gave detailed, constructive feedback to help us improve our work, while acknowledging the future potential of our approach.
>
> We are currently working on additional experiments that compare LF's performance against ANGC, and evaluate LF's performance on MNIST and CIFAR-10 to further support its robustness across tasks.
>
> In the meantime, we want to address the other concerns and questions the reviewer raised:
>
> - **Biological insights beyond just "it can solve MinAtar."**
> Thanks for raising this, we're happy for this opportunity to elaborate. Since this issue also provides useful context for our other responses, we'll address it first.
>
>   Our motivation for designing LF is heavily biologically inspired. We saw a potential connection between the design principles of Forward-Forward (FF) and biological reward-based learning via striatic dopamine release; LF is our attempt to synthesize the two to improve deep Q-learning.
>
>   FF provides an alternative to backpropagation that is more biologically plausible and computationally efficient, because it does not suffer from weight transport or update locking; Ororbia and Kifer (2022) also discusses these issues. Like LF, FF globally broadcasts the target to each layer of the network, which each computes their own local loss and updates. However, FF does not provide a biologically plausible explanation on how the target signal can be globally broadcasted in the brain. Interestingly, recent research in neuroscience by Li and Jasanoff (2020) found that in the striatum, dopamine release can be a global signal that induces local changes to reward stimulation, providing a potential answer in the case of reward-based learning. Additionally, biological reward-based learning has close ties with TD-learning, which is used as an explanatory model for dopamine-induced synaptic modifications in the striatum (Starkweather and Uchida, 2021). The alignment between FF's principle of globally-induced local learning, and evidence of biological globally-induced local modifications for reward-based learning in the brain, is the central biological insight that motivated us to develop LF as a Q-learning algorithm.
>
>   To be clear, we do not believe that LF plausibly represents the complete reward-learning process in the striatum. Unlike backpropagation, the brain likely utilizes multiple types of learning in conjunction, both supervised and unsupervised, using local and global signals (O'Reilly et al., 2012). LF only models one type of learning, i.e. error-driven learning using global reward signals to induce local updates. Other biologically plausible algorithms, like ANGC that the reviewers cited, may provide explanations to other types of biological learning. Indeed, aspects of Hebbian learning or active inference are likely critical to achieving a full biologically plausible, efficient, and powerful learning system.  In that light, we examine LF here not to replace other methods, but to demonstrate that this one principle alone is sufficient to be able to solve some complex RL tasks nearly as well as backprop. We believe that a key to achieving general, human-like intelligence will be the integration of these different learning methods and learning signals; this is an exciting direction we aim to explore in future work.
>
> - **Background on the forward-forward algorithm is insufficient.**
> We expanded the background section on FF to include more mathematical/algorithmic details. This will be reflected in the upcoming updated manuscript. Specifically, the first paragraph of Section 2.2 now reads:
> "The Forward-Forward (FF) algorithm is a greedy multi-layer learning algorithm that replaces the forward and backward passes of backpropagation with two identical forward passes, positive and negative.
> These passes differ only in the data fed through the network and the objective they optimize. For a positive pass, the update increases the goodness $g$ of that layer.
> While Hinton proposes several measures of goodness, the de facto measure used is the sum of each squared hidden activation $h_{i}$ of the layer minus a threshold $\theta$, i.e. $g = \sum\_{i}{h\_{i}^2} - \theta$.
> In the case of a negative pass, the update decreases goodness.
> The positive data are real examples while the negative data is `fake.' One way to generate the fake data is by masking and overlaying real examples.
> For the MNIST dataset, this could be combining the top half of a 7 with the bottom of a 6.
> The goal of the model is to correctly classify an input as either positive (real) or negative (fake).
> To optimize each layer's weights, FF minimizes the binary cross-entropy loss between the sigmoid of the goodness $\sigma(g)$ and the ground truth."

---

> > ### Author Response · Authors · 2023-11-18
> >
> > - **The reward signal is used globally, despite the claim that we do not propagate loss signal between cells.**
> > The reviewer is correct-- however, sending the reward globally is actually intended, and is biologically motivated by known processes from reward-based learning in the striatum (as discussed in our response to the biological insights question).
> > Although global propagation of loss/error signals is not biologically plausible, broadcasting a reward signal globally is plausible (Li and Jasanoff, 2020).
> >
> >   We understand this can likely be a common point of confusion. To improve clarity, we updated Section 3.3, it now reads: "Different from standard deep Q-learning however, each layer of the network computes its own local loss and update using a global reward signal. Specifically, the reward is broadcasted to all cells, and each cell computes their own Q-value prediction, TD-target, and TD-loss for learning. Globally broadcasting the reward signal to enable local computation of loss and learning is biologically motivated, imitating reward-learning processes in the striatum, which globally broadcasts reward signals via dopamine release that induces local updates (Li and Jasanoff, 2020)."
> >
> > - **MDPs can be handled by simple feedforward NNs, so what is the design motivation for the recurrent-like connections of LF cells?**
> > There are two main benefits of adding the forward connections in time (which are the recurrent-like connections in our architecture).
> > First, they provide the agent a prior about the temporal structure of the Q-value trajectories, which improves performance.
> > Feedforward NNs certainly have the capacity to solve MDPs, and approximate the true Q-function of any policy arbitrarily well given enough neurons and the correct parameters.
> > However, this doesn't mean that feedforward NNs are great at learning these parameters from trajectories.
> > We can draw a similar analogy in computer vision.
> > In theory, all image classification tasks that can be handled by CNNs can also be handled by feedforward NNs.
> > But, we prefer using CNNs because the weight-sharing in convolutional layers provide a prior about the spatial structure of natural images, which helps models learn more efficiently.
> > In the case of Q-learning, our insight is this:
> > given a good policy, the Q-value predictions of a well-trained model
> > should remain stable through each state of a trajectory (assuming the dynamics
> > are reasonably deterministic).
> > This means that the Q-value prediction of the current timestep, and the activations used to compute them, are often still useful for computing the prediction for the next timestep.
> > This is a useful property of the temporal structure of Q-value trajectories that we want to leverage for more efficient learning; thus, we pass the hidden activations forward in time.
> >
> >   Second, the forward-in-time connections allow us to pass hidden activations downwards through the network while avoiding circular dependencies. Since we do not backprop error signals, the forward connections serve as the sole information channel for upper layers to communicate with lower layers. As our experiments show, having this information channel is critical to performance on more complex tasks. For example, on Seaquest, the LF agent without forward connections barely achieves any return, but with the connections, LF nears the performance of DQN. However, if we naively pass activations downwards in the same timestep, we run into a circular dependency issue-- the activations of layers depend on each other. While it's sometimes possible to resolve this by allowing the activations to circulate through the network until they reach a fixed-point, these circulations are computationally inefficient and not biologically-plausible (Bartunov et al, 2018). By passing the activations down forwards in time, we cleanly avoid this issue.

---

> ### Author Response · Authors · 2023-11-18
>
> - **Computational cost of LF compared to DQN.**
> The computational efficiency gains of LF over DQN come from two main sources: increased potential for parallelization, and decreased data movement costs.
> For DQNs (and backpropagation in general), because the error of each lower layer is dependent on higher layers, the updates must be computed sequentially layer-by-layer.
> In contrast, since LF computes losses and updates locally (using a global signal), the layers of LF are not dependent on each other, which allows the learning to be parallelized across the layers.
> Futhermore, DQNs suffers from the weight transport problem, which is not only biologically implausible, but also significantly increases its computational efficiency on neuromorphic hardware (Crafton et al., 2019).
> For full transparency, we want to clarify that these efficiency gains directly result from the design principles that we inherited from Forward-Forward, and have already been raised by Hinton (2022); achieving these efficiencies over backpropagation is not a novel contribution of our work.
> Our contribution here lies in proposing a new algorithm that extends these efficiency improvements to deep Q-learning, which is notoriously computation-heavy.
>
>   While each layer does use more neurons and connections to generate their own Q-value predictions, we took this into consideration when selecting our network sizes-- this is why each layer of the LF network has less hidden activations. In addition, since LF does not suffer from update locking, the Q-value predictions of each layer can be performed in parallel. More specifically, in terms of number of parameters, in our experiments, we used a similar number of parameters for both the LF network and the DQN to make the performance comparisons more fair and direct. Since the MinAtar testbed slightly varies in input channels and action space across the environments, these numbers also vary depending on the environment. Given an environment with 4 input channels and 3 actions, the LF network had 1,199,619 trainable parameters, and the DQN had 1,123,200. We added the calculations of these numbers to the Appendix, which will be reflected in the updated manuscript.
>
> **References**
>
> [1] Li, Nan, and Alan Jasanoff. "Local and global consequences of reward-evoked striatal dopamine release." Nature 580.7802 (2020): 239-244.
>
> [2] Bartunov, Sergey, et al. "Assessing the scalability of biologically-motivated deep learning algorithms and architectures." Advances in neural information processing systems 31 (2018).
>
> [3] Crafton, Brian, et al. "Local learning in RRAM neural networks with sparse direct feedback alignment." 2019 IEEE/ACM International Symposium on Low Power Electronics and Design (ISLPED). IEEE, 2019.
>
> [4] Hinton, Geoffrey. "The forward-forward algorithm: Some preliminary investigations." arXiv preprint arXiv:2212.13345 (2022).
>
> [5] O'Reilly, Randall C., et al. Computational cognitive neuroscience. Vol. 1124. Mainz: PediaPress, 2012.
>
> [6] Ororbia A, and Kifer D. The neural coding framework for learning generative models[J]. Nature communications, 2022, 13(1): 2064.
>
> [7] Starkweather, Clara Kwon, and Naoshige Uchida. ”Dopamine signals as
> temporal difference errors: recent advances.” Current Opinion in Neurobiology
> 67 (2021): 95-105.

---

> ### Author Response · Authors · 2023-11-23
>
> Hey again Reviewer AyQt,
>
> Just wanted to give you an update re: the additional experiments we promised to run-- they're done and in the updated manuscript :)
>
> In Appendix C we added new experiments on 4 new RL environments, Cart Pole, Mountain Car, Lunar Lander, and Acrobot to further show the applicability of our algorithm across different RL environments, and indirectly compare LF's performance against ANGC.
> In Appendix D we added results on MNIST and CIFAR-10. LF's performance across all these tasks look strong!
>
> We chose Cart Pole, Mountain Car, Lunar Lander, and Acrobot as our new environments because they are similar to those used by ANGC (Ororbia and Mali, 2022), which is the closest work to ours.
> We reached out to the authors for code of their agent, but unfortunately they needed more time to prepare their code for public release, which they said they will likely complete in December.
> In light of this, rather than comparing ANGC against LF in the MinAtar environments we used, we evaluated LF in the 3 publicly available environments that Ororbia and Mali used to evaluate ANGC, as well as a similar environment Acrobot-v1.
> To indirectly compare our performance against ANGC, we reimplemented the DQNs that ANGC benchmarked against, and use their performance as a reference point (we also compare against CleanRL's DQN for good measure).
> As a pleasant surprise, LF handily outperforms the DQN in all 4 environments.
> Its training process is in general more stable, and reaches higher average return on each environment.
>
> For MNIST and CIFAR-10, LF also performed reasonably well, achieving 98.74% test accuracy on MNIST, and 57.25 on CIFAR-10. We also have results for different network sizes, check out our new Appendix D.
>
> In addition, we thought that your question on biological insights is very important, and agree that we didn't do LF justice on this matter in our writing-- so we added parts of our response to your question to Appendix E.
>
> Lastly, we added Ororbia and Mali (2022) and Ororbia and Kifer (2022)  to our related work section. They are closely relevant to our work, thanks for bringing them to our attention.
>
> Once again, thank you for the detailed feedback to help us improve our work!

---

### Official Review · Reviewer_MymM · 2023-10-25

**Soundness:** 2 fair
**Presentation:** 3 good
**Contribution:** 2 fair
**Rating:** 3
**Confidence:** 4

**Summary:**

This work employed a recently proposed algorithm known as the Forward-Forward algorithm, instead of Back-Propagation, to train deep neural networks in the context of reinforcement learning tasks.

**Strengths:**

The author has clearly delineated both the work already accomplished and the work that remains to be done.

**Weaknesses:**

The main drawback of the article lies in its lack of substantial content. Specifically, the experimental content is minimal, lacks robustness, and lacks detail or novelty. Since this paper does not introduce any novel methods, and the author simply applies the existing FF algorithm to RL tasks, it is essential for the author to thoroughly evaluate the differences between FF and BP algorithms in the context of RL tasks. In terms of experiments, there are several areas that need improvement:

1. FF algorithm - There are multiple variations of the FF algorithm, and the author should consider testing more than one rather than relying solely on a single FF algorithm.

2. RL algorithms - Beyond just Q-learning, the author should explore other RL algorithms to provide a more comprehensive analysis.

3. RL tasks - The paper only covers five basic tasks, but it would be beneficial for the author to expand their analysis to a wider range of tasks to effectively compare FF and BP.

In the analysis, the current focus is primarily on performance. However, it would be beneficial for the author to explore and compare other aspects impacted by FF and BP algorithms, thus enhancing the depth of the article.

**Questions:**

Section 2.1, the sixth line, this formula is written incorrectly.

---

> ### Author Response · Authors · 2023-11-18
>
> We thank the reviewer for their comments and suggestions.
>
> Unfortunately, we believe there are some significant misunderstandings, particularly regarding the novelty of our algorithm and our contributions, that may have affected the reviewer’s judgment. This likely reflects that our paper does not present these aspects of our work sufficiently clearly. Thus, we have adjusted Sections 1,  2 and 3 of our upcoming updated manuscript accordingly to better communicate them, but we also want to first provide a more direct response here.
>
> - We believe the reviewer misinterpreted our work as just an application of the existing FF algorithm.
> In the weaknesses section, the reviewer states that we “simply [apply] the existing FF algorithm to RL tasks.” We want to clarify that this is not true— one of our central contributions is that we introduce a novel learning algorithm for Q-learning.
> The novelty of our method is also supported by the review of Reviewer AyQt.
> Our algorithm builds upon FF, inheriting the local computation of losses and forward-in-time connections to propagate information downwards, but differ substantially to adapt it to our domain.
>
>    Unlike FF, our algorithm does not make 2 forward passes through the network during training. We only make a single forward pass, then provide the reward globally to every cell, such that they can each locally compute their own TD error to update their parameters. Rather than comparing positive and negative data, like FF, the comparison is between the model’s current prediction of the “goodness” of its situation, and its updated prediction in the next timestep. In some sense, the models’ current prediction *is* the negative data, and the TD error entices it to generate positive data, i.e. the true Q-value given the current policy and environment— this is why we don’t need 2 separate forward passes. The local computation of loss using a global target is indeed like FF. However, FF is designed for classification tasks, whereas Q-learning is a regression task. To solve our task, we introduced new architecture in the form of our LF cells, which utilize an attention-like mechanism to handle regression and normalization of the activity vectors, as we show in Figure 2 and Section 3.1 of our paper. We hope this explanation addresses some of the concerns regarding **Weakness 1**; rather than testing existing variations of FF, we built upon it to design our own new version that is more suited to our domain, and showed that it achieves comparable performance to DQN in the MinAtar testbed.
>
> - We also want to address **Weakness 2**, and explain why we chose to work on Q-learning rather than explore more broadly applying variations of FF to RL methods.
> There are two interesting synergies between the TD-learning setting and the properties of FF we inherited; they motivated us to develop LF specifically for Q-learning.
> First, it has a biologically plausible explanation.
> TD-learning has strong ties to reward-based learning in the brain, and is used as an explanatory model for synaptic modifications in the striatum (Starkweather and Uchida, 2021).
> Importantly, research suggests that the striatum uses dopamine release to globally broadcast signals that affect response to reward stimulation (Li and Jasanoff, 2020).
> LF's use of global reward signals to modulate local learning of Q-values imitates this process.
> This also answers a question that FF did not explain: how would a biological NN send the global target/label to every layer?
> For reward-based learning, the brain could be broadcasting the global target via dopamine.
> Thus, framing LF as a TD-learning algorithm is the most biologically plausible.
>
>   Second, the forward connections in time are well-suited to take advantage of the temporal structure of the Q-values of trajectories for better learning. Given a good policy, the Q-value predictions of a well-trained model should remain stable through each state of a trajectory (assuming the dynamics are reasonably deterministic). This means that the Q-value prediction of the current timestep, and the hidden activations used to make this prediction, can often still be useful for predicting the Q-value of the next timestep. In contrast, in FF's experiments on image classification, this effect is forced-- to ensure that the activations of the current timestep is still useful for the next, FF has to repeat the same input every timestep, reducing computational efficiency. Our results empirically support the particular effectiveness of forward connections for Q-learning; they qualitatively improve performance in the more complex MinAtar environments.

---

> ### Author Response · Authors · 2023-11-18
>
> - We have a two-part answer regarding **Weakness 3**, i.e. by conducting our experiments on MinAtar, we only cover 5 basic tasks.
> First, while solving MinAtar might not be challenging for more established deep RL algorithms, in our experience they are not certainly not basic tasks for a novel algorithm that does not utilize backpropagation.
> For example, Active Neural Generative Coding (Ororbia and Mali, 2022), another recently developed biologically-inspired deep Q-learning algorithm, is evaluated on four simpler environments, but their work is nonetheless a very insightful contribution that progresses the field towards stronger biologically-plausible learning algorithms.
> Thus, we believe that most deep RL researchers, if told that an algorithm can perform as well as backprop on MinAtar, without using backprop to coordinate learning across the layers, would find it at least interesting, if not unexpected.
>
>   Second, previous research has shown that experiments in the smaller-scale MinAtar environment can still yield valuable insights for deep RL. Obando-Ceron and Castro (2021) showed that most results achieved by the Rainbow paper (Hessel et al., 2018) in the Atari environments can be replicated in the MinAtar environments. Therefore, they argue that Hessel et al. would likely have arrived at the same qualitative conclusions if they conducted their experiments in MinAtar. Additionally, they urge the field to change the status-quo of excessively focusing on large-scale tasks, and avoid dismissing empirical work that focus on smaller problems, as this reduces barriers of entry for newcomers from underprivileged communities, which we fully support. Given the potential of LF (also supported by the review of Reviewer AyQt), we believe that its strong performance on MinAtar is a valuable result, and would thus like to share it with the community.
>
> - Re: your concern raised in the Questions section, we removed the misplaced $\pi(a|s)$, thank you for pointing it out.
>
>
>
> **References**
>
> [1] Starkweather, Clara Kwon, and Naoshige Uchida. "Dopamine signals as temporal difference errors: recent advances." Current Opinion in Neurobiology 67 (2021): 95-105.
>
> [2] Li, Nan, and Alan Jasanoff. "Local and global consequences of reward-evoked striatal dopamine release." Nature 580.7802 (2020): 239-244.
>
> [3] Ororbia, Alexander G., and Ankur Mali. "Backprop-free reinforcement learning with active neural generative coding." Proceedings of the AAAI Conference on Artificial Intelligence. Vol. 36. No. 1. 2022.
>
> [4] Obando-Ceron, Johan Samir, and Pablo Samuel Castro. "Revisiting rainbow: Promoting more insightful and inclusive deep reinforcement learning research." International Conference on Machine Learning. PMLR, 2021.
>
> [5] Hessel, Matteo, et al. "Rainbow: Combining improvements in deep reinforcement learning." Proceedings of the AAAI conference on artificial intelligence. Vol. 32. No. 1. 2018.

---

> > ### Comment · Reviewer_MymM · 2023-11-22
> >
> > Thanks for the detailed explanation, which change my view on the novelty and contribution part. I have raised the score accordingly. But I still hope to see more experiments.

---

> > > ### Author Response · Authors · 2023-11-23
> > >
> > > Thanks for the response! We have updated the manuscript, and included more experiments on 4 new RL environments, Cart Pole, Mountain Car, Lunar Lander, and Acrobot. We chose these environments because they were used to evaluate another recent biologically-inspired algorithm for deep Q-learning, Active Neural Generative Coding (Ororbia and Mali, 2022). We also added experiments on MNIST and CIFAR-10, following Reviewer AyQt's suggestion. We're very glad to report that LF performs well on all of the new experiments. Please check out our new Appendix C and Appendix D, and hope this helps quell your remaining concerns!

---

### Official Review · Reviewer_eRt9 · 2023-11-01

**Soundness:** 3 good
**Presentation:** 4 excellent
**Contribution:** 3 good
**Rating:** 6
**Confidence:** 2

**Summary:**

The paper describes "Local-Forward," a method that combines an ad-hoc neural architecture (L-F cells) and a reinforcement learning strategy, overall avoiding backpropagation of the gradients.

**Strengths:**

The paper is well-written, and the presentation is optimal. As a general comment on the experimental results, I think they are well-described and well-commented.
I also particularly appreciated the "Limitations" section.

**Weaknesses:**

I'm not familiar with RL; hence, I urge the AC to take this into consideration to appropriately weigh my review. My remarks on the methods could be wrong or trivial. That being said, I will try to provide only minor comments on some aspects I believe might improve the quality of this work.

1. I would avoid making claims that are hard, if not impossible, to substantiate, such as the one in the first bullet point at the end of the introduction: "We propose an alternative to backpropagation for reinforcement learning that does not suffer contradictions with known biology." I suggest making this statement somewhat less definitive.

2. You initially define the value function with the symbol $\mathcal{Q}$ at the beginning of section 2.1,  but then you go on to use the symbol $Q$. Moreover in the recursive relation for $Q$ some lines below in the lhs you have $Q^\pi$ while in the rhs you dropped the "policy" superscript. Is this intentional?

3. I would spend some more words to describe who are and how the matrices $W_{\rm in}$ and $W_{\rm query}$ act; probably inserting them also in Figure~2 would help.

4. Conventionally the "argmax" indicates a set (of all the points I which the function at hands assume minimal value), but in all your formula,  you use it as if it were a real number. I would suggest to either clarify this or consider using a different notation. For instance is Algorithm 1 I would write $a_t\in {\rm arg max} \dots$ rather than $a_t = {\rm arg max} \dots$

**Questions:**

How do you explain that in some experiments, it is possible to achieve decent results without forward connections? If I've understood correctly, aren't the forward connections the ones that provide an information signal from higher layers?

---

> ### Author Response · Authors · 2023-11-18
>
> We thank the reviewer for their comments and suggestions.
>
> We particularly appreciate that the reviewer took extra time to give specific, actionable feedback. We agree with the weaknesses, and addressed each concern in our upcoming updated manuscript to improve it. Specifically:
>
> - We modulated our claim in the first bullet point of the contributions; it now reads “We propose an alternative to backpropagation for reinforcement learning that suffers less contradictions with known biology."
> - We standardized our use of the $Q$ notation in the background section.
> - We rewrote our description of the cell internals to better explain the roles of the matrices $W\_{in}$ and $W\_{query}$. We experimented with adding $W_{in}$ and $W_{query}$ to Figure 2, but since they live within the ReLU and tanh weight layers, we felt that it might distract from the high-level structure the Figure aims to convey.
>     Instead, in the new text, we included a reference to the parts of Figure 2 the matrices correspond to; let us know what you think.
>     The paragraph now reads: "We introduce an attention mechanism to compute the Q-value prediction.
>     Each cell first computes its hidden activation, $h^{[l]}\_t$, using the layer above's hidden activations from the previous timestep, $h\_{t-1}^{[l+1]}$, and the layer below's hidden activations at the current timestep, $h\_{t}^{[l-1]}$.
>     Specifically, it passes the concatenation of $h\_{t-1}^{[l+1]}$ and $h\_{t}^{[l-1]}$ through a ReLU weight layer (shown in Figure~2) to get $h^{[l]}\_t$.
>     The ReLU weight layer multiplies the concatenated input $[h^{[l-1]}\_t, h^{[l+1]}\_{t-1}]$ by its learned weight matrix $W^{[l]}\_\text{in}$, then applies the ReLU nonlinearity function.
>     Next, the cell passes $h^{[l]}\_t$ through the tanh weight layers, each corresponding to an action $a$, which multiplies $h^{[l]}\_t$ by its learned weight matrix $W^{[l]}\_\text{query}$, then applies the tanh function.
>     The output of the tanh layers plays a role akin to attention weights, although they do not have to sum to 1.
>     Finally, the cell takes the inner product between the output of the tanh layers, $\text{tanh}(W^{[l]}\_\text{query} \cdot h^{[l]}\_t)$, and the hidden activations $h^{[l]}\_t$, to compute the cell's Q-value prediction
>     $\hat{Q}^{[l]}(s\_t,a)$.
>     This structure is heavily inspired by the successful attention architecture, and allows each cell to attend to the incoming information dynamically per action."
> - We changed the first usage of argmax in Algorithm 1 to $a_{t} \in argmax$, and the rewrote the second usage with a max function instead to prevent this confusion.

---

> ### Author Response · Authors · 2023-11-18
>
> Regarding your question: this is indeed an interesting and surprising observation— intuitively, it seems that a multi-layer network without an information channel from upper to lower layers should not perform well, as the upper layers cannot provide any feedback to the lower layers.
> We can’t claim any credit for the discovery of this unexpected behavior; it was first raised by Hinton (2022).
> Hinton showed that, for the MNIST classification task, the lower layers of a multi-layer network trained without any downward connections can nonetheless learn useful features for upper layers, and thus achieve good classification performance.
>
> But as our experiments suggest, the importance of downward information channels increases with the difficulty of the task.
> In our results, the performance of the 3-layer network without forward connections is strongest on Freeway, where its average episodic return rivals that of LF and the backprop-trained DQN.
> This is likely because Freeway is the simplest of the MinAtar tasks, and unlike the other environments, it has an upper limit on episodic return, which bounds performance.
> In the other 4 environments, the episode length is determined by when the agent fails and dies, and the difficulty of the game increases as time goes on.
> In contrast, Freeway has a fixed episode length, and does not increase in difficulty: the goal for Freeway is to cross the road as many times as possible within a fixed 2500-frame window.
> But, even with the optimal solution, the maximum possible number of times an agent can cross the road within that window is 50-60, depending on how lucky the agent gets with the traffic patterns, which is random.
> Therefore, the reason that the 3-layer network without forward connections catches up to LF on Freeway is likely not because it is as good, but because LF has no more room to improve its performance.
> On the other hand, on Seaquest, where agents must learn a complex, multi-step rescue-and-resurface procedure to achieve higher returns, we can observe the opposite behavior.
> Both the 3-layer network without forward connections and the single-layer network are unable to learn the rescue-and-resurface procedure, and as a result sit flatly near 0 return throughout training.
>
> **References**
>
> [1] Geoffrey Hinton. The forward-forward algorithm: Some preliminary investigations. arXiv
> preprint arXiv:2212.13345, 2022.

---

### Author Response · Authors · 2023-11-23

Dear Reviewers,

We uploaded a new version of our manuscript. Following each reviewer’s advice, the major changes are as follows:

- We added new experimental results for 4 additional environments, Cart Pole, Mountain Car, Lunar Lander, and Acrobot, in Appendix C. We selected these environments because they were used to evaluate another recent biologically-inspired learning algorithm for deep RL, Active Neural Generative Coding (Ororbia and Mali, 2022). We show that LF achieves strong performance in all 4 environments. (Reviewer AyQt, Reviewer MymM)
- We added new experimental results on MNIST and CIFAR-10 in Appendix D. (Reviewer AyQt)
- We updated the Introduction, Background, and Method sections to more clearly describe the contributions and novelty of our work, and improved the precision of our claims. (Reviewer MymM, Reviewer eRt9)
- We updated our Related Works section to include the relevant literature raised by Reviewer AyQt.
- We added a new section further discussing the biological connections of LF, in Appendix E. (Reviewer AyQt)
- We added calculations for the number of trainable parameters LF uses, in comparison to the DQN in our experiments, in Appendix B. (Reviewer AyQt)
- We fixed a coding error that we previously reported in our Limitations section, which caused performance to precipitously crash in specific edge cases.


We hope these updates, together with our responses, addresses your concerns.
We truly believe that Local Forward shows exciting potential, and would love to share it with the community.
Thank you for your time and feedback!

**Reference**

Ororbia, Alexander G., and Ankur Mali. "Backprop-free reinforcement learning with active neural generative coding." Proceedings of the AAAI Conference on Artificial Intelligence. Vol. 36. No. 1. 2022.

---

### Meta-Review · Area_Chair_FTe5 · 2023-12-11

**Metareview:**

This submission proposes "Local-Forward," an new temporal-difference learning algorithm for prediction Q-values. This algorithm attempts to replace the biologically-implausible backpropagation algorithm with a method that utilizes local updates and forward connections, drawing inspiration from the Forward-Forward algorithm and integrating aspects of recurrence and attention in neural architectures. This is an important topic, and the paper went through major updates during the rebuttal.

However, despite the authors' efforts to clarify, the core novelty of the proposed Local-Forward algorithm remains ambiguous. Reviewers pointed out that it appears to be a variant of the existing Forward-Forward algorithm applied to reinforcement learning tasks, and the authors' response, while detailed, has not fully addressed these concerns. The scope and depth of the experiments conducted to validate the proposed algorithm are limited. Although the authors have expanded their experimental setup in response to reviewer feedback, the additional experiments still do not comprehensively demonstrate the algorithm's effectiveness or superiority over existing methods. While the paper aims to address the biological implausibility of backpropagation, it does not convincingly establish the biological plausibility of its own approach. See reviewer comments below for full details of these points.

**Justification For Why Not Higher Score:**

Given these concerns, especially around novelty, experimental rigor, and biological plausibility, the paper does not meet the high standards expected for acceptance at the ICLR conference.

**Justification For Why Not Lower Score:**

N/A

---

### Decision · Program_Chairs · 2024-01-16

Reject